**Peer**J

# Phylogenetic congruence of lichenised fungi and algae is affected by spatial scale and taxonomic diversity

Hannah L. Buckley, Arash Rafat, Johnathon D. Ridden,
Robert H. Cruickshank, Hayley J. Ridgway and Adrian M. Paterson

Department of Ecology, Lincoln University, Lincoln, Canterbury, New Zealand

Corresponding author
Hannah L. Buckley,
Hannah.Buckley@lincoln.ac.nz

## ABSTRACT

The role of species' interactions in structuring biological communities remains unclear. Mutualistic symbioses, involving close positive interactions between two distinct organismal lineages, provide an excellent means to explore the roles of both evolutionary and ecological processes in determining how positive interactions affect community structure. In this study, we investigate patterns of co-diversification between fungi and algae for a range of New Zealand lichens at the community, genus, and species levels and explore explanations for possible patterns related to spatial scale and pattern, taxonomic diversity of the lichens considered, and the level sampling replication. We assembled six independent datasets to compare patterns in phylogenetic congruence with varied spatial extent of sampling, taxonomic diversity and level of specimen replication. For each dataset, we used the DNA sequences from the ITS regions of both the fungal and algal genomes from lichen specimens to produce genetic distance matrices. Phylogenetic congruence between fungi and algae was quantified using distance-based redundancy analysis and we used geographic distance matrices in Moran's eigenvector mapping and variance partitioning to evaluate the effects of spatial variation on the quantification of phylogenetic congruence. Phylogenetic congruence was highly significant for all datasets and a large proportion of variance in both algal and fungal genetic distances was explained by partner genetic variation. Spatial variables, primarily at large and intermediate scales, were also important for explaining genetic diversity patterns in all datasets. Interestingly, spatial structuring was stronger for fungal than algal genetic variation. As the spatial extent of the samples increased, so too did the proportion of explained variation that was shared between the spatial variables and the partners' genetic variation. Different lichen taxa showed some variation in their phylogenetic congruence and spatial genetic patterns and where greater sample replication was used, the amount of variation explained by partner genetic variation increased. Our results suggest that the phylogenetic congruence pattern, at least at small spatial scales, is likely due to reciprocal co-adaptation or co-dispersal. However, the detection of these patterns varies among different lichen taxa, across spatial scales and with different levels of sample replication. This work provides insight into the complexities faced in determining how evolutionary and ecological processes may interact to generate diversity in symbiotic association patterns at the population and community levels. Further, it highlights the critical importance of considering sample replication, taxonomic diversity and spatial scale in designing studies of co-diversification.

## INTRODUCTION

Ecologists still do not have a full understanding of how species interactions affect the structure of biological communities. We have a long history of theoretical and empirical work on the roles of competition and predation (*Chase & Leibold, 2003*; *Tilman, 1982*), but a much poorer understanding of how positive interactions, such as facilitation and mutualisms, drive community phenomena, such as species diversity (*Gross, 2008*; *Stachowicz, 2001*). Over the last few decades, there has been an increase in interest in the role of positive interactions, with many empirical studies showing that it is the balance between positive and negative interactions that is important in structuring communities (e.g., *Bertness & Callaway, 1994*; *Elias et al., 2008*; *LaJeunesse, 2002*; *Thrall et al., 2007*; *Waterman et al., 2011*).

Because mutualistic symbioses involve very close positive interactions between two distinct organismal lineages, they provide an excellent opportunity to specifically explore how positive interactions influence community structure and to evaluate the relative importance of evolutionary and ecological processes in the way that positive interactions affect community structure. Tightly interacting taxa, such as host–parasite systems and certain plants and their insect pollinators, typically show high degrees of phylogenetic congruence between hosts and associates, where the phylogeny of one taxon closely tracks the phylogeny of the partner taxon (*Cuthill & Charleston*; *Light & Hafner, 2008*; *Quek et al., 2004*; *Subbotin et al., 2004*), largely due to co-evolutionary processes. What is not as clear is whether most obligate species-level symbiotic relationships, such as seen in lichens and corals, also have measureable levels of codivergence. Typically, these diffuse symbiotic and mutualistic interactions are complicated by variation in partner identity or where more than one symbiotic partner is involved, such as vascular plants and their root-inhabiting mutualists (*Hollants et al., 2013*; *Lanterbecq, Rouse & Eeckhaut, 2010*; *Walker et al., 2014*). Further, it is much less clear that the causal processes involved in these coevolving symbiotic relationships will produce a pattern of codivergence given the increased opportunity for host switching.

Lichens are a classic example of a mutualistic symbiosis. Lichen thalli are the result of an association between a fungus (the mycobiont) and a photobiont, which is usually a green alga, but may also be a cyanobacterium (*Nash, 2008*). Around 3–4% of lichens are tripartite involving a symbiosis of both a green alga and a cyanobacterium (*Henskens, Green & Wilkins, 2012*). In all lichens, the photobiont provides photosynthate to the mycobiont, which in turn provides habitat, water and nutrients to the photobiont (*Honegger, 1991*; *Nash, 2008*). The symbionts of lichens are relatively poorly known because they are very often difficult to culture and identify, particularly many of the photobiont partners (*Ahmadjian, 1993*; *del Campo et al., 2013*; *Grube & Muggia, 2010*;

*Honegger, 1991*). However, in the last two decades a great deal of molecular work has been done to address this, showing variable patterns in partner identity and other patterns of association (*DePriest, 2004*). This variable, and seemingly diffuse, mutualism provides a complex model system for addressing questions regarding the role of evolutionary processes in forming and driving ecological patterns in species interactions and how this affects community structure.

The first step in understanding how a mutualistic symbiosis might affect community structure is to determine whether or not there is specificity in the symbiosis. In the case of lichens, we know that although there is no evidence for very tight co-evolutionary relationships at the species level, phylogenetic patterns have been observed where particular fungal taxa preferentially partner with particular algal taxa (*Fernández-Mendoza et al., 2011*; *Yahr, Vilgalys & Depriest, 2004*) resulting in a correlation in the respective genetic distances of each partner. For example, *Widmer et al. (2012)* found similar genetic structures for the lichen symbiosis between the mycobiont, *Lobaria pulmonaria*, and its photobiont, *Dictyochoropsis reticulata* within Europe. Conversely, it seems that some lichenised algal taxa are capable of partnering with a range of fungal taxa (*Beck, 1999*); thus, the specificity of the symbiosis appears to be driven by fungal selectivity (*sensu Beck, Kasalicky & Rambold, 2002*). For example, *Ruprecht, Brunauer & Printzen (2012)* observed that Antarctic lecideoid lichens were not specific for particular algae, except for two fungal species, which preferentially associated with a particular algal clade within *Trebouxia* sp. Thus, it appears that there is variability in association patterns among different lichen taxa (*Fahselt, 2008*).

Several mechanisms are proposed to underpin the patterns in phylogenetic congruence observed for lichens. First, co-evolutionary processes, whereby one partner adapts to take better advantage of the symbiosis, may lead to a reciprocal adaptive evolutionary change in the other partner, although there is little evidence for this in the literature (*Yahr, Vilgalys & DePriest, 2006*). Second, many lichens asexually reproduce, either by fragmentation or specialised structures (*Walser, 2004*), so that the resulting offspring lichens contain clones of their parents, otherwise known as 'vertical transmission' (*Dal Grande et al., 2012*; *Werth & Scheidegger, 2011*). Sexual reproduction in green algal photobionts (apart from those in the Trentepholiales) is thought to be extremely rare within lichen thalli (*Friedl & Büdel, 2008*; *Sanders, 2005*), despite evidence of recombination within these taxa (*Kroken & Taylor, 2000*). If symbiont co-dispersal is coupled with genetic drift, a pattern of co-diversification is likely to emerge. However, 'horizontal transmission' of photobionts into newly forming thalli is thought to occur, such as in the form of escaped zoospores (*Beck, Friedl & Rambold, 1998*), and population genetics studies show evidence of algal switching (*Dal Grande et al., 2012*; *Kroken & Taylor, 2000*; *Nelsen & Gargas, 2008*; *Piercey-Normore & DePriest, 2001*). Further, most green algal photobionts commonly occur in a free-living state, although for some taxa, such as *Trebouxia*, much about their life cycles and availability in the environment is unknown (*Sanders, 2005*). A free-living state is likely to decrease the importance of vertical transmission and decrease the congruence of phylogenetic patterns. Third, spatial structure in fungal and algal distributions could

drive patterns in phylogenetic congruence. Spatial structure in either fungi or algae could arise through dispersal limitation, habitat fragmentation, or niche differentiation, such as variation in habitat preferences. For example, *Werth et al. (2007)* showed that the mycobiont of the cyanobacterial lichen, *Lobaria pulmonaria*, varied genetically over spatial scales of less than a few kilometres in a pasture-woodland landscape and suggested that this could be caused by dispersal limitation among habitat patches. *Peksa & Škaloud (2011)* showed that the spatial distribution patterns of *Asterochloris* in Europe and North America, the green algal photobiont for two different lichen genera, were driven by substrate type and relative exposure to rain and sun. If spatial structure in fungal and algal distributions resulted in limited availability of one or both partners relative to the other, this would lead to a congruent phylogenetic pattern. For example, *Marini, Nascimbene & Nimis (2011)* found that communities of epiphytic lichens with different photobiont types (Chlorococcoid green algae, Cyanobacteria or *Trentepohlia*) showed different biogeographic patterns across climatically different areas within Italy. Thus, if spatial structure in genetic variation results from differential distributions of algal or fungal ecotypes, this could result in phylogenetic congruence for specimens compared across environmental gradients. For example, the patterns in variable algal selectivity that *Vargas Castillo & Beck (2012)* observed within the genus *Caloplaca* in the Atacama Desert in Northern Chile were related to changing habitat conditions along an altitudinal gradient.

Although much recent research has shown that lichenised fungi specialise on particular algae regardless of the availability of other species, most work has been conducted at the within-species and within-genus levels, and much less often at higher levels of phylogenetic diversity. Such patterns and their explanations, like most ecological phenomena, are likely to be scale-dependent and related to both small scale processes, such as the dispersal of lichen propagules, as well as larger scale biogeographic processes and climatic variation. In addition, these patterns are likely to depend on the amount of phylogenetic diversity contained within the dataset considered. We expect that if co-evolutionary processes play a role, then phylogenetic congruence should be stronger when considering higher levels of phylogenetic diversity because they are the accumulation of a longer period of evolutionary change. To examine these patterns and effects, we tested patterns of association for a range of New Zealand lichens at the community, genus, and species levels. We assembled six independent datasets that varied in the spatial extent of sampling, taxonomic diversity and the level of specimen replication so that we could compare patterns in phylogenetic congruence across these variables. We have taken a novel approach to the analysis of phylogenetic congruence that uses Moran's eigenvector mapping, distance-based redundancy analysis and variance partitioning, which allows us to evaluate the effects of sampling on the quantification of phylogenetic congruence. We interpret the patterns in the light of the relative importance of the mechanisms driving variation in co-diversification patterns.

**Peer**J

**Table 1 List of the datasets analysed showing the number of specimens sampled, the approximate number of lichen morphotypes, and the number of sites sampled.** Also given are the maximum distance between two sample points (Spatial extent) and the mean (standard deviation) genetic distance for each matrix. Note that only 28 of the 58 Flock Hill multiple species dataset specimens were mapped and were therefore analysed separately.

| Dataset | Number of specimens | Taxonomic variation | Number of sites | Spatial extent (m) | Algal genetic diversity | | Fungal genetic diversity | |
|---|---|---|---|---|---|---|---|---|
| | | | | | Mean (S.D.) | Range | Mean (S.D.) | Range |
| NZ *Ramalina* | 21 | Few morphotypes (3) | 9 | 581,576 | 0.08 (0.05) | 0.0–0.16 | 0.06 (0.04) | 0.0–0.11 |
| NZ *Usnea* | 111 | Many morphotypes (17) | 43 | 1,251,276 | 0.09 (0.06) | 0.0–0.18 | 0.05 (0.02) | 0.0–0.10 |
| NZ *Usnea* replicated | 83 | Several morphotypes (9) | 18 | 882,910 | 0.10 (0.06) | 0.0–0.17 | 0.05 (0.02) | 0.0–0.09 |
| Craigieburn *Usnea* | 36 | Several morphotypes (6) | 1 | 1,775 | 0.04 (0.04) | 0.0–0.15 | 0.03 (0.02) | 0.0–0.06 |
| Flock Hill *Usnea* | 66 | Few morphotypes (3) | 1 | 1,095 | 0.03 (0.02) | 0.0–0.14 | 0.03 (0.02) | 0.0–0.08 |
| Flock Hill community mapped | 28 | Many lichen genera | 1 | 796 | 0.05 (0.04) | 0.0–0.18 | 0.11 (0.07) | 0.0–0.29 |
| Flock Hill community | 58 | Many lichen genera | 1 | 1,141 | 0.08 (0.07) | 0.0–0.49 | 0.14 (0.06) | 0.0–0.30 |

## METHODS

### Lichen specimen collection

Lichen thallus samples were collected from many mapped locations around both the North and South Islands of New Zealand (under New Zealand Department of Conservation low-impact research and collection permit, CA-31641-FLO). Samples were taken either from the ground, or from trees and structures like fence posts and stored in paper envelopes. Each sample was identified to the lowest taxonomic level possible and assigned a specific sample code. Specimens are held in collections at Lincoln University. Five non-overlapping sample sets were collected (Table 1): (1) Multiple lichen species collected on mountain beech (*Nothofagus solandri* var. *cliffortioides*) trees within less than 1 km$^2$ of Flock Hill Station (*Buckley, 2011*), (2) *Usnea* spp. specimens from Flock Hill Station, (3) *Usnea* spp. specimens from Craigieburn Forest Park, (4) *Usnea* spp. specimens from sites around New Zealand, and (5) *Ramalina* spp. specimens from around New Zealand (Fig. 1).

### Molecular analysis

Total DNA was extracted from surface sterilised lichens using the Plant DNA Mini Kit (Bioline, London, UK) following the manufacturer's instructions. Photobiont and mycobiont ITS rRNA were amplified using specific algal (nr-SSU-1780-5′ Algal and ITS4) and fungal (nr-SSU-1780-5′ Fungal and ITS) primers described by *Piercey-Normore & DePriest (2001)*. PCR amplification was performed in a 25 µl reaction volume. Each 25 µl reaction contained 1 × GoTaq® Green Master Mix, 5 pmol of each primer, 10 µg purified bovine serum albumin (BSA; New England BioLabs, Ipswich, MA, USA) and 1 µl of the extracted DNA (25–30 µg/µl). The thermal cycle for the algal reaction was as follows: initial denaturation at 94 °C for 2 min, then 35 cycles of denaturation at 94 °C for 30 s, annealing at 50 °C for 45 s, extension at 72 °C for 2 min, then a final extension at 72 °C

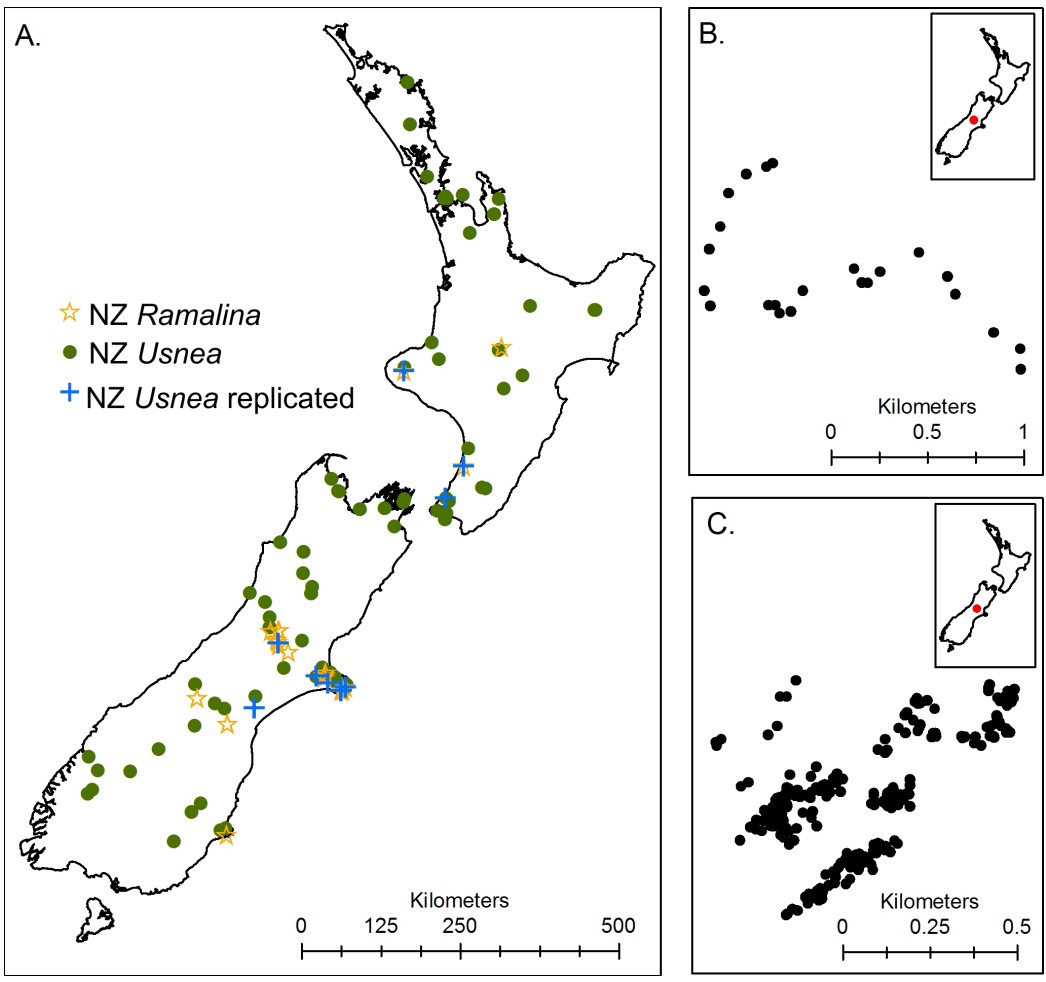

**Figure 1 Maps showing sample collection locations for (A) the three New Zealand datasets, (B) the Craigieburn *Usnea* dataset and (C) the Flock Hill community and *Usnea* dataset.** Note that the Craigieburn samples were collected along a road running along an elevation gradient.

for 7 min. The thermal cycle for fungal rRNA amplification was the same as the algal one except the annealing time was increased to 54 °C. Sequences were deposited in GenBank. We used a GenBank BLAST search of sequences from this dataset to estimate the number of fungal operational taxonomic units (OTUs) as 19 (see Table S1). In all datasets, including the *Usnea* and *Ramalina* datasets, algal sequences were matched to GenBank sequences associated with green algae in the genus *Trebouxia* or closely related genera.

### Data analysis

For each of the six datasets, algal and fungal DNA sequences were aligned separately using 'Prankster' (*Löytynoja & Goldman, 2005*) with the default parameters. These alignments were used to calculate genetic distance matrices (see Table S2) from the raw distances using uncorrected p-distances (*Paradis, 2006*) implemented by the 'dist.dna' function in the 'ape' package in R (*R Core Team, 2013*). We repeated the analyses using a genetic distance matrix calculated using the TN93 substitution model (*Tamura & Nei, 1993*). We also repeated

these analyses using patristic distances derived from a Bayesian phylogenetic analysis to enable us to compare this 'tree-free' method to one based on a full phylogenetic analysis. Bayesian trees were calculated using a lognormal molecular clock in BEAST v1.8. We used these results to calculate patristic distances (sum of branch lengths) from the maximum likelihood tree (calculated in MEGA v.6.0) using the R package 'adephylo'. The results from both alternative analyses (not shown) were congruent with those obtained using p-distances, so we present the p-distance results only.

To relate fungal and algal genetic distances to each other and to describe their variation in space, distance matrices were used in a combined analysis using Moran's eigenvector mapping and variance partitioning (*Borcard, Gillert & Legendre, 2011*, pp. 258). This analysis was used to describe and partition the variation in algal genetic distances between (a) the fungal genetic distance matrix and (b) a matrix of spatial variables. The spatial variables were derived using the Moran's eigenvector maps (MEMs) procedure (*Borcard, Gillert & Legendre, 2011*) implemented using the 'pcnm' function in the R package 'vegan', which uses a principal coordinates analysis to represent different scales of spatial variation for the given set of sample locations (*Borcard, Gillert & Legendre, 2011*). MEM analysis produces one fewer spatial variables than there are sample points, describing all possible spatial variation in the data from broad scale variation to very fine scale variation. Only the most important subset of these spatial variables (MEMs) was included in a distance-based redundancy analysis (db-RDA) analysis, which relates multivariate data (algal genetic distance matrix) to explanatory matrices (fungal genetic distance matrix and spatial variables). The selected MEMs were those that were significantly related to the algal distance matrix in a distance-based RDA and forward selection procedure using the 'capscale' and 'ordistep' functions in the 'vegan' package in R (*Oksanen et al., 2013*). Variance partitioning calculations were conducted following procedures outlined in *Borcard, Gillert & Legendre (2011)*. The 'capscale' function performs a redundancy analysis that seeks the series of linear combinations of the explanatory factors that best describe variation in the response matrix, constrained by the two explanatory matrices (*Borcard, Gillert & Legendre, 2011*). The variance partitioning procedure computes $R^2$ canonical values analogous to the adjusted $R^2$ values produced in multiple regression (*Peres-Neto et al., 2006*). The analysis indicates how much total variation in the response matrix (e.g., algal genetic distance) is explained by each of the explanatory matrices alone, as well as the component of shared variation, e.g., spatially structured variation in fungal genetic distances. This analysis was also performed using the fungal genetic distance matrix as the dependent matrix and the algal genetic distance matrix as the explanatory matrix to allow comparison of the degree of spatial correlation in each of the matrices.

To test the significance of the phylogenetic congruence between fungi and algae for each of the six datasets, we used the Procrustes approach to co-phylogeny (PACo, *Balbuena, Míguez-Lozano & Blasco-Costa, 2012*). This procedure performs a principal coordinates analysis on the algal genetic distance matrix followed by a Procrustes rotation of the fungal genetic distance matrix, while retaining the information that algae and fungi are paired in particular lichen specimens (*Balbuena, Míguez-Lozano & Blasco-Costa, 2012*). A sum

of squares is calculated from the individual residuals for each specimen that represents the lack of fit of the fungal genetic distance matrix onto the principal coordinate analysis result for the algal genetic distance matrix (*Balbuena, Míguez-Lozano & Blasco-Costa, 2012*). The algal–fungal pairing matrix, i.e., which alga is paired with which fungus, is then randomised 10,000 times and the sums of square values recalculated. The observed sum of squares value is then compared to the distribution of values from the randomisations to determine the probability of obtaining the observed result under random expectation (*Balbuena, Míguez-Lozano & Blasco-Costa, 2012*). The magnitude of the residual for each lichen specimen shows its relative lack of fit to a co-diversification pattern. Therefore, for three datasets for which we had additional information on specimen traits, we compared individual residuals among specimens to determine which ones contributed most to the observed association pattern. These three datasets (and trait information) were the Flock Hill community (growth form), Flock Hill *Usnea* and New Zealand *Usnea* datasets (apothecia present or absent). Raw genetic and geographic distance matrices are provided in Tables S1 and S2.

## RESULTS

The six datasets captured a wide range of geographic extent and, unsurprisingly, the datasets with greatest numbers of different lichen morphotypes contained the greatest fungal genetic diversities (Table 1); fungal genetic diversity was strongly correlated with the spatial extent of sampling (Pearson's $r = 0.91$; $n = 6$). However, algal genetic diversity was not correlated with fungal genetic diversity (Pearson's $r = 0.10$; $n = 6$) or with the spatial extent of sampling (Pearson's $r = -0.10$; $n = 6$).

The db-RDA showed that spatial variables were important for explaining fungal and algal genetic diversity patterns in all datasets (Table 2, Significant MEMs). Large-scale variables, i.e., low numbered MEMs, were important for all spatially-structured datasets. Genetic variation in *Ramalina* fungi and their associated algae was significantly related to several large-scale MEMs, showing that spatial pattern in relatedness varied at the larger scales within this spatial extent, such as between the North and South Islands (Table 2). Some intermediate scale MEMs were also important, illustrating additional, more complex, spatial patterns. Similarly, for the *Usnea* datasets, large-scale MEMs were of greatest importance, along with some intermediate-scale, but no very fine-scale, MEMs. Some intermediate-scale MEMs were important in explaining algal and fungal variation in the Flock Hill community dataset (Table 2).

Variance partitioning showed that a large proportion of variance in both the algal and fungal genetic variation was explained by genetic variation in the partner (Fig. 2). In general, the proportion of explained variation was high, at 75% or more (Table 2, Fig. 2). Interestingly, across all datasets for which spatial variation was important, the fungal genetic variation was better explained by spatial variables than was the algal genetic variation for the same lichens (Fig. 2). As the spatial extent of the samples increased, so too did the proportion of explained variation that was shared between the spatial variables and the partners' genetic variation (Fig. 2).

**Table 2 Results from db-RDA with variance partitioning and Procrustes approach to co-phylogeny giving the *P*-value from a test randomising the association matrix for fungi and algae for the six independent datasets and the full Flock Hill community dataset.** Note that only 28 of the 58 Flock Hill multiple species dataset specimens were mapped and were therefore analysed separately from the full dataset. Variance partitioning divides the total variance up into portions explained by the partner genetic distance matrix (Partner), the purely spatial portion (Space), the spatially structured variation in the partner matrix (Shared) and unexplained variation (Unexpl.). The significant MEMs are given in order of their significance. For each dataset, there are $n-1$ MEMs in the total set and smaller MEM numbers represent larger-scale spatial pattern.

| Dataset | Significant MEMs | Partner | Shared | Space | Unexpl. | *P* |
|---|---|---|---|---|---|---|
| **Dependent matrix: algae** | | | | | | |
| NZ *Ramalina* | 1, 5, 3, 2, 7 | 0.21 | 0.65 | 0.09 | 0.06 | 0.016 |
| NZ *Usnea* | 1, 4, 6, 12, 17, 7, 3, 9, 2, 5, 45, 8, 63, 38, 41, 28, 54 | 0.39 | 0.41 | 0.08 | 0.12 | <0.001 |
| NZ *Usnea* replicated | 4, 5, 1, 3 | 0.89 | 0.07 | 0.00 | 0.04 | <0.001 |
| Craigieburn *Usnea* | 2, 9 | 0.87 | 0.00 | 0.01 | 0.12 | 0.021 |
| Flock Hill *Usnea* | 16 | 0.62 | 0.01 | 0.01 | 0.35 | 0.030 |
| Flock Hill community mapped | 9, 5 | 0.64 | 0.08 | 0.11 | 0.12 | <0.001 |
| Flock Hill community | – | 0.65 | – | – | 0.35 | <0.001 |
| **Dependent matrix: fungi** | | | | | | |
| NZ *Ramalina* | 1, 3, 5, 7, 9, 10, 2 | 0.32 | 0.62 | 0.02 | 0.03 | 0.025 |
| NZ *Usnea* | 1, 4, 12, 3, 8, 6, 9, 45, 17, 54, 13, 38, 41, 16, 5, 56, 7 | 0.45 | 0.23 | 0.12 | 0.2 | <0.001 |
| NZ *Usnea* replicated | 4, 3, 31, 1, 17, 5, 9 | 0.72 | 0.01 | 0.02 | 0.25 | <0.001 |
| Craigieburn *Usnea* | 7, 12, 13, 15, 16, 3 | 0.55 | 0.02 | 0.3 | 0.14 | 0.024 |
| Flock Hill *Usnea* | 1 | 0.62 | 0.00 | 0.08 | 0.30 | 0.022 |
| Flock Hill community mapped | 11, 1 | 0.61 | 0.05 | 0.09 | 0.25 | 0.002 |
| Flock Hill community | – | 0.66 | – | – | 0.34 | <0.001 |

PACo analysis showed that the co-diversification for all datasets, regardless of whether the dependent matrix was the algae or the fungi, was significant at the alpha = 0.05 level (Table 1), indicating that it is very unlikely that the correlations in the fungal and algal genetic distances have arisen by chance. When comparing the individual residuals from the PACo analysis, for the Flock Hill community dataset, the *Usnea* and *Ramalina* specimens, which made up all of the fruticose specimens, had relatively good fit to the co-diversification pattern, compared to most other taxa (Fig. 3). For the Flock Hill *Usnea* and New Zealand *Usnea* datasets, the presence of apothecia did not appear to be related to the relative specimen contribution to the lack of phylogenetic congruence (Fig. 3).

## DISCUSSION

We show that there is a strong and significant pattern of phylogenetic congruence for the algae and fungi found in New Zealand lichens, both across multiple taxa within a community and within two widespread genera sampled at spatial extents from less than 1 km$^2$, to the whole country (*c.* 260,000 km$^2$). For instance, at the small spatial scales of the Flock Hill study site, there was surprisingly high phylogenetic congruence, as demonstrated by the high variance explained by partner genetic variation. Such congruence suggests that fungi and algae are not randomly distributed across the lichen symbiosis for the taxa considered. Our results show that this non-random pattern is

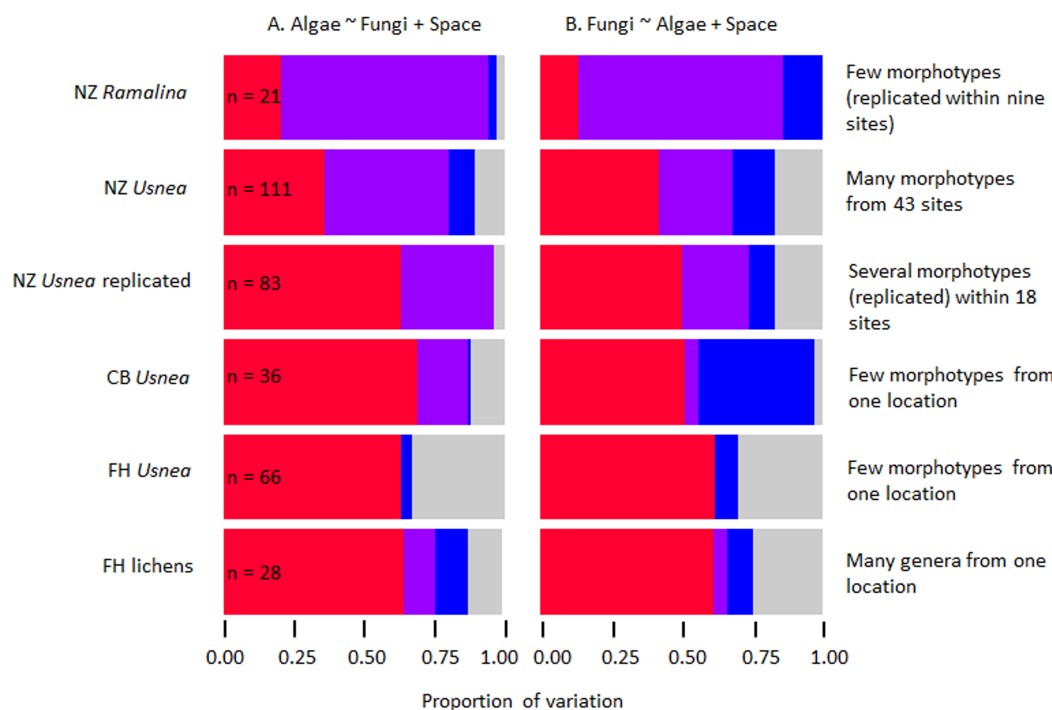

**Figure 2** **Bar charts showing variance partitioning for six independent datasets modelled as algal genetic variance as a function of fungal genetic variance and spatial variation (A) and fungal genetic variance as a function of algal genetic variance and spatial variation (B).** The total variation in genetic distance is explained by partner genetic distance (red), independent spatial variation (blue), and spatially-structured variation in partner genetic distances (purple). Unexplained variation is shown in grey. The number of specimens sampled (*n*) is given for each dataset.

affected by (1) spatial scale, (2) taxon considered, (3) taxonomic diversity and (4) level of sample replication.

The amount of genetic variation explained by spatially-structured partner genetic variation increased with increasing spatial extent (Fig. 2) suggesting that a large amount of phylogenetic congruence is likely to be due to the distributions of fungi and algae at larger, rather than smaller, spatial scales. In addition, the spatial structuring of *Usnea* and *Ramalina* fungal genetic distances was more prominent than for algae, suggesting that drivers of fungal distributions were more important in determining these congruence patterns than drivers of algal distributions. It appears that the algae are more dependent on the distribution of the fungi than the fungi are on the algae. Thus, the strong spatial signal in our results shows that at large spatial scales, and consequently larger taxonomic scales in the case of the four *Usnea* datasets, at least some of the co-diversification pattern appears likely to be due to processes other than co-evolution or vertical transmission. If factors such as variation in habitat preferences or photobiont availability were important, we would expect to see an increase in the proportions of genetic variance being explained by spatially-structured variation in the genetic distances of the partners. This is indeed the result we observe across the four *Usnea* datasets. One other paper that mentions the effects of spatial scale on phylogenetic congruence patterns is a study of *Cladonia* lichens across

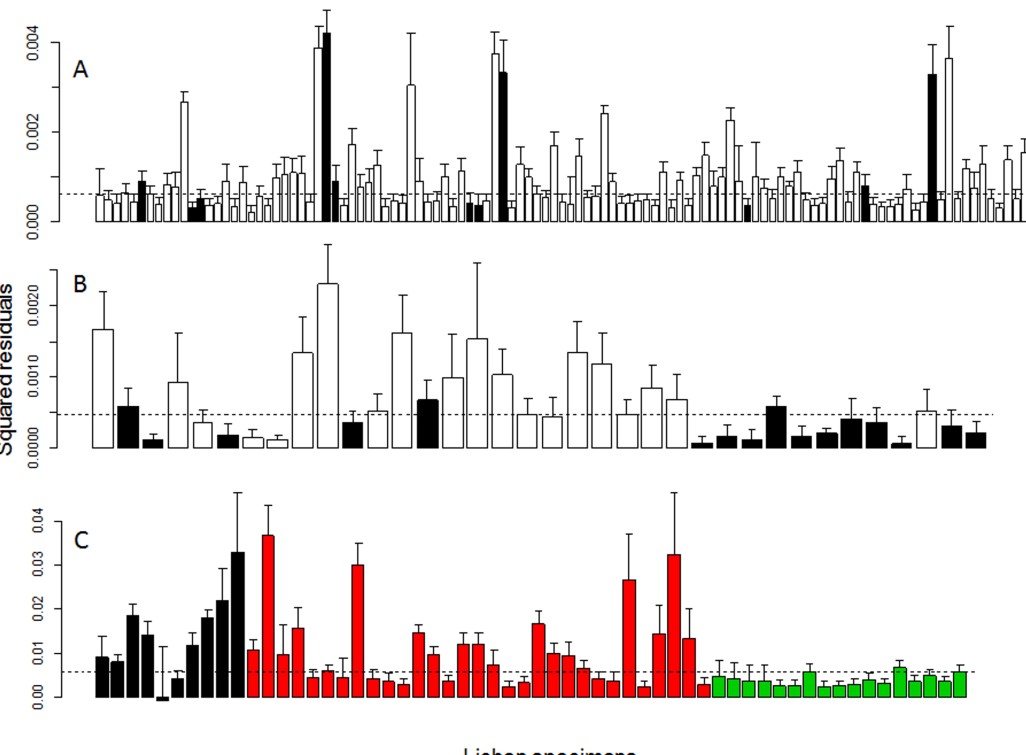

**Figure 3 Bar charts showing the individual lichen specimen contribution to the Procrustes sums of squares for (A) the New Zealand *Usnea* dataset ($n = 111$), (B) the Craigieburn *Usnea* dataset ($n = 36$) and (C) the Flock Hill all specimen dataset ($n = 58$).** Dashed line indicates the median sums of squares value. Bars for (C) are coloured by growth form: crustose (black), foliose (red) and fruticose (green) and for the other two datasets black bars indicate specimens that had apothecia and white bars are those that were asexual. Note that in (C) all but the first from the left of the fruticose specimens were specimens of *Usnea* or *Ramalina*.

six discrete rosemary scrub sites in three regions in Florida (*Yahr, Vilgalys & Depriest, 2004*). Their findings illustrated that, across Florida, photobiont genetic variation was not significantly spatially structured, despite different fungi occurring at different sites. These contrasting findings suggest that different patterns of phylogenetic congruence may occur in different lichen taxa.

The variation in the patterns from the PACo for individual lichen specimens also suggest that the taxon considered may affect the observed phylogentic congruence pattern. The fruticose taxa had lower residual values in the PACo analysis showing that they contributed relatively more to the phylogentic congruence pattern than did crustose and foliose taxa. However, these lichen specimens were all *Usnea* and *Ramalina* suggesting that this variation may have a phylogenetic basis, rather than being due to the life form. There was little pattern observed when we related the presence of apothecia to their contribution to the co-diversification pattern for the *Usnea*-only datasets (Fig. 3) suggesting that, at least at this scale and level of replication, ability for the fungal component to sexually reproduce had little to do with the observed pattern. Residual values are not comparable among datasets, so it is not possible to evaluate these differences among spatial scales. Overall,

these results are consistent with the variable patterns in lichen co-diversification in the literature (*Ruprecht, Brunauer & Printzen, 2012*; *Vargas Castillo & Beck, 2012*).

Our study encompassed a range of taxonomic diversities for both fungi and algae. Specifically, when fungal diversity was very high, algal diversity was also relatively high, despite the very small spatial scale (Table 1, Flock Hill community dataset). For *Usnea*, when fungal genetic diversity was low, algal diversity was low or high depending on whether the spatial scale was small or large, respectively (Table 1). However, if we consider only the two datasets at the smallest spatial scale, despite the variation in genetic diversity, the pattern of phylogenetic congruence did not vary; the Flock Hill datasets both show high levels of phylogenetic congruence despite having the largest difference in genetic diversity (Fig. 2). This is consistent with arguments suggesting that coevolution is not an important driver of the lichen symbiosis (*Yahr, Vilgalys & DePriest, 2006*), because if coevolution was important, then we would expect to see an increase in the degree of phylogenetic congruence with increasing genetic diversity.

By contrasting the result from the NZ *Usnea* dataset and the NZ *Usnea* replicated dataset, we can consider the effects of within-site replication on the phylogenetic congruence signal. This result shows that where greater replication was used, the amount of variation explained by partner genetic variation increased. This highlights the importance of considering sampling design in studies of phylogenetic congruence.

This work leads us to more questions regarding variation in phylogenetic congruence patterns and its causes, which are likely to be scale-dependent. We need better understanding of the factors influencing association patterns including algal availability and niche differentiation, dispersal (metapopulation dynamics), and reproductive traits. In particular, we need to understand the effects of lichen reproductive modes better as many lichens reproduce both sexually and asexually. When the fungal component sexually reproduces, the symbiosis must re-form which gives the fungus an opportunity to change symbiotic partners. With asexual reproduction, the fungus and alga disperse together, and so, a predominance of asexual reproduction may be one of the reasons we see such a strong co-diversification signal in these taxa. However, the results of this study suggest that the spatial distributions of the fungi and algae may also be important in determining the nature of the symbiosis, particularly at larger spatial scales, as has been observed in some parasite lineages (e.g., *du Toit et al., 2013*). Thus, we need to do more work on algal and fungal availability to determine if habitat preferences and/or dispersal limitation led to some of the spatial patterns that we see. In addition, we need better understanding of the availability of free living algae to lichens and what their microhabitat preferences are.

The analyses used in this study do not require phylogenetic trees. The advantage of not requiring phylogenetic trees is that we avoid computationally intensive methods when generating distance matrices, but arrive at the same conclusions in this case. The disadvantage of not using phylogenetic trees is that the results cannot be placed explicitly in a phylogenetic context denying the opportunity to reconstruct individual evolutionary events, such as algal switches among fungal lineages. However, these global analysis methods provide a broad picture of codiversification patterns, which is a fair, and possibly

more accurate, reflection of the diffuse nature of the lichen symbiosis. Specifically, this study is consistent with previous research showing that, despite the diffuse nature of the lichen mutualistic symbiosis, there is strong selectivity within the association. We show that despite spatial structuring in algal, and particularly fungal, distributions at large spatial scales, the phylogenetic congruence pattern, at least at small spatial scales, is due to either reciprocal co-adaptation or, more likely, to co-dispersal. However, the influence of these processes is likely to differ among different lichen taxa. This work gives us insight into some of the complexities we face in determining how evolutionary and ecological processes may interact to generate diversity in symbiotic association patterns at the population and community levels.

## ACKNOWLEDGEMENTS

The authors would like to thank: Elizabeth Bargh, Jennifer Bannister, Alison Knight, Mike Bowie, John Marris, Dan Blachon, Brad Case and Sam Case for specimen collection; Ursula Brandes, Hamish Maule, Ben Myles and Natalie Scott for field assistance; Richard Hill for land access permission; Ben Myles, David Galloway, Jennifer Bannister, and Alison Knight for lichen identification; Dalin Brown, Seelan Baskarathevan and Chantal Probst for assistance with molecular analysis; Norma Merrick for DNA sequencing; Brad Case for GIS work; the Lincoln University Spatial Ecology and Molecular Ecology Groups for discussion; and Richard Cowling, Terry Hedderson and one anonymous reviewer for suggestions that greatly improved this manuscript.

### Funding

Funding for this project was provided by Lincoln University, the Brian Mason Scientific and Technical Trust, the Canterbury Botanical Society, and the Bayer Boost Scholarships programme. The funders had no role in study design, data collection and analysis, decision to publish, or preparation of the manuscript.

### Grant Disclosures

The following grant information was disclosed by the authors:
Lincoln University.
The Brian Mason Scientific and Technical Trust.
The Canterbury Botanical Society.
The Bayer Boost Scholarships programme.

### Competing Interests

Hannah L. Buckley is an Academic Editor for PeerJ.

### Author Contributions

- Hannah L. Buckley conceived and designed the experiments, performed the experiments, analyzed the data, contributed reagents/materials/analysis tools, wrote the paper, prepared figures and/or tables, reviewed drafts of the paper.

- Arash Rafat conceived and designed the experiments, performed the experiments, analyzed the data, wrote the paper, reviewed drafts of the paper.
- Johnathon D. Ridden performed the experiments, analyzed the data, wrote the paper, reviewed drafts of the paper.
- Robert H. Cruickshank conceived and designed the experiments, analyzed the data, contributed reagents/materials/analysis tools, wrote the paper, reviewed drafts of the paper.
- Hayley J. Ridgway conceived and designed the experiments, performed the experiments, contributed reagents/materials/analysis tools, wrote the paper, prepared figures and/or tables, reviewed drafts of the paper.
- Adrian M. Paterson wrote the paper, reviewed drafts of the paper.

### Field Study Permissions

The following information was supplied relating to field study approvals (i.e., approving body and any reference numbers):

Field collections were approved by the New Zealand Department of Conservation under a low-impact research and collection permit (CA-31641-FLO).

### Supplemental Information

Supplemental information for this article can be found online at http://dx.doi.org/10.7717/peerj.573#supplemental-information.

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
