# Peer review of "Phylogenetic congruence of lichenised fungi and algae is affected by spatial scale and taxonomic diversity"

_PeerJ, doi:10.7717/peerj.573_

## Round 0.1 · original submission · Major Revisions

Both reviewers find your article interesting and worthy of publication. However, both believe there are problems of clarity that require attention before the article is acceptable for publication in PeerJ. In particular, the use of genetic distance as a surrogate for formal phylogenetic analysis requires better justification.

Reviewer 1 ·

Basic reporting

As best I can tell, there is no discussion of where the collections from this study were deposited (Herbarium). This would be especially useful if someone later wants to identify the collections here.

And I also can find no mention of where the DNA sequences are deposited. There is a lot of really good data in this study, but I really think it should be made publicly available (I believe this is also a requirement of PeerJ).

Experimental design

The authors pose some very interesting questions, and collect and analyze a large amount of data with sophisticated methods, that permit them to tease apart the influence of different variables on the patterns of association. The general approach taken is a bit unorthodox (when comparing with other lichen studies), in the sense that there is no discussion of the taxonomic/phylogenetic affinities of the fungal/algal organisms in the community dataset. For it to work (and I think they can make it work), I feel the authors need to clarify some things, specifically, (1) what levels of taxonomic diversity are encompassed in the community datasets (this is sort of touched on in Table 1 in the sense that the max/min and mean genetic variation was given for the entire sample), (2) were Usnea/Ramalina morphotypes a good proxy for phylogenetic relatedness/distinctness (ie. were sequences within individual morphotypes monophyletic?)…perhaps a phylogeny would be a useful way to present this? Essentially, the morphotypes were treated as phylogenetic units when sampling, but it is unclear if this translates to what is observed in the sequence data.

The authors collected a number of lichens at various sites. In some replicates, these were limited to species of a specific fungal genus (Usnea in some, Ramalina in others), while in the community-level study, lichens (regardless of fungal or algal taxonomy) were collected and sequenced. What is difficult is to understand the phylogenetic scope (fungal and algal) that is encompassed within community-level study. I feel this is important because it seems as though symbiont switching coupled with codiversification could produce the same results as observed here (instead of just codiversification), and (all else being equal) I'd expect symbiont switches to increase with time. For instance, in the community level analyses, the authors could have studied collections from three separate fungal orders, which associate with very different algae. Under this scenario, the closely related fungi would associate with a similar pool of algae but these fungal and algal symbionts are all very different from one another. Now suppose these fungi or algae were derived from different lichenization events (evolutionarily) - this pattern of similar fungi associating with similar algae would not necessarily be due strictly to codiversification (since the MRCA of these two algal lineages was not lichenized, and these algal lineages separately formed evolutionary associations with algae). Placing this all in an evolutionary (co-diversification) context is especially challenging as their is very little discussion of what organisms were investigated in the community data set. The lack of discussion in the context of phylogenetic scale makes it challenging to assess the validity of the conclusions.

In the community level analysis, it might be useful to provide a supplementary table with a listing of what fungal families or genera these samples were from - or else a list of the best (or few best) BLAST hits (coverage/identity) could instead be provided - that might at least narrow the identity of each the sample down (even if it's only to a coarse level/scale). It would also be useful to know roughly what sorts of algal clades were sampled…I imagine Trebouxia for the Usnea/Ramalina specimens (you don't have to mention what species, but it would be useful to state/verify that Trebouxia was the photobiont recovered). For the community dataset, it also would be useful to list close BLAST hits for the algal sequences in a supplementary table.

Validity of the findings

This is certainly an interesting study with a lot of data, and some great analyses. I think it will make a very nice paper, and a great contribution, but some points need to first be clarified (see above) before the validity of all the findings can be fully assessed.

Additional comments

p1: typo - couple of words run together "levelsampling", "spatialvariables", "congruencepattern"

P2, L26: I'd say it's an issue of both selectivity and specificity and not just specificity.

L50-51: see also Beck et al. (2002) New Phytologist 153: 317-326 for nice discussion on selectivity and specificity in lichen associations.

L52-53: This sentence doesn't really seem to follow the previous - it goes from discussing selectivity to lack of phylogenetic congruence and then to variability among associations. Re-work this part.

L53-54: See Yahr et al. (2004), also.

L184-185: clarify if this statement refers to both fungi and algae.

L186-187: I'd change Ramalina algae to Ramalina-associated algae, or Ramalina fungi and their associated algae.

L203: change foliose to fruticose.

L221-224: These two sentences are discussing different things, and I guess I don't see how the present study finds the opposite of Marini et al (at least in the way it is presented in the present paper)? The authors state that Marini et al. found species level diversity of fungi was dependent on the distribution of broad algal groups, while the present study is suggesting that algal distribution depends on fungal distribution, and does not discuss fungal species richness vs. algal distribution.

L226: "…large spatial scales…" and phylogenetic scales, too, right?

·

Basic reporting

Overall I find the article to be clear and well-written, with the exception of a few awkward sentences and a a few ambiguities. I have indicated these on the annotated ms attached.

The Introduction seems to me to be overly long and could probably be trimmed by 20% without undue loss of important background information.

Experimental design

A key aim of the paper is to measure phylogenetic congruence between the fungal and photobiont components of lichens at a range of spatial scales. However, there is no phylogeny reconstruction per se - rather genetic distances are used as a surrogate for phylogeny. In some (perhaps even many) cases this is perfectly OK as rates of evolution do not vary wildly across samples and lineages. In the present instance I have no way of evaluating whether this is the case for these data sets. The authors need to provide an explicit justification for conflating the two in this particular instance. If there are problems, then the analyses need to be redone using "proper" tree-based analyses (e.g. formal measures of PD between samples)

The paper also claims to test the effects of phylogenetic diversity. This term has come to have a quite specific meaning in the evolutionary literature (i.e. the sum of branch lengths connecting a given set of samples/taxa) and I don't believe that this is what the authors intend here. Rather phylogenetic diversity seems to refer to species diversity??? This needs to be clarified.

Validity of the findings

Given the caveats stated above, I believe the findings are valid. If, however, there is strong heterogeneity in evolutionary rates then many of the statistics may be strongly biased by rare, strongly deviating, samples.

Additional reporting would go some way to alleviating these fears. For example, it would be useful to know something about the diversity of ITS haplotypes in each data set, along with summary measures of things like nucleotide diversity between haplotypes (or even summary of the p-distances that are so crucial to the analyses). Similarly I would like to see at least a brief summary of how diverse the photobiont community is.

Additional comments

No comments

---

## Round 0.2 · accepted · Accept

I am satisfied with the responses to the reviewer's comments. Indeed, these have been addressed comprehensively and cogently.